# Measuring single cell divisions in human tissues from multi-region sequencing data

Benjamin Werner [1,2✉], Jack Case[1,3], Marc J. Williams [4,5,6], Ketevan Chkhaidze[1], Daniel Temko[4], Javier Fernández-Mateos[1], George D. Cresswell [1], Daniel Nichol[1], William Cross[4], Inmaculada Spiteri[1], Weini Huang[7,8], Ian P.M. Tomlinson [9], Chris P. Barnes [5,10], Trevor A. Graham [4✉] & Andrea Sottoriva [1✉]

Both normal tissue development and cancer growth are driven by a branching process of cell division and mutation accumulation that leads to intra-tissue genetic heterogeneity. However, quantifying somatic evolution in humans remains challenging. Here, we show that multi-sample genomic data from a single time point of normal and cancer tissues contains information on single-cell divisions. We present a new theoretical framework that, applied to whole-genome sequencing data of healthy tissue and cancer, allows inferring the mutation rate and the cell survival/death rate per division. On average, we found that cells accumulate 1.14 mutations per cell division in healthy haematopoiesis and 1.37 mutations per division in brain development. In both tissues, cell survival was maximal during early development. Analysis of 131 biopsies from 16 tumours showed 4 to 100 times increased mutation rates compared to healthy development and substantial inter-patient variation of cell survival/ death rates.

[1] Evolutionary Genomics and Modelling Lab, Centre for Evolution and Cancer, The Institute of Cancer Research, London, UK. [2] Evolutionary Dynamics Group, Centre for Cancer Genomics & Computational Biology, Barts Cancer Institute, Queen Mary University of London, Charterhouse Square, London EC1M 6BQ, UK. [3] University of Cambridge, Cambridge, UK. [4] Evolution and Cancer Laboratory, Centre for Cancer Genomics & Computational Biology, Barts Cancer Institute, Queen Mary University London, London, Charterhouse Square, London EC1M 6BQ, UK. [5] Department of Cell and Developmental Biology, University College London, London, UK. [6] Centre for Mathematics and Physics in the Life Sciences and Experimental Biology (CoMPLEX), University College London, London, UK. [7] Group of Theoretical Biology, The State Key Laboratory of Biocontrol, School of Life Science, Sun Yat-sen University, 510060 Guangzhou, China. [8] School of Mathematical Sciences, Queen Mary University London, London, UK. [9] Institute of Cancer and Genomic Sciences, University of Birmingham, Birmingham, UK. [10] UCL Genetics Institute, University College London, London, UK. ✉email: b.werner@qmul.ac.uk; t.graham@qmul.ac.uk; andrea.sottoriva@icr.ac.uk

Most cells in human tissues have a limited life span and need to be replenished for tissues to remain functional[1–3]. This cell turnover leads to somatic evolution, with cells accumulating mutations upon which selection may act[4,5]. Inter- and intra-tumour genetic heterogeneity[6,7] as well as treatment resistance[8,9] are now understood to be consequences of somatic evolutionary processes. Recent studies demonstrate somatic evolution in healthy non-cancerous tissues throughout live[10–14]. Normal brain cells carry hundreds of mutations weeks after conception[12] and normal skin or esophagus cells accumulate hundreds of cancer driver mutations during adulthood[10,11].

These observations call for a better quantitative understanding of the somatic evolutionary forces in both cancerous and healthy tissues[15]. However, unlike species evolution, for which a timed fossil record exists[16,17], the lack of sequential human data over time due to ethical and technical limitations is a major obstacle. Furthermore, some evolutionary forces are difficult to measure. For example, the mutational burden in a tissue is the combined effect of per-cell mutation and per-cell survival rates, which remain hidden in sequencing data[18,19] (Fig. 1). Currently, we cannot independently infer these two for somatic evolution fundamental quantities from single time point sequencing data.

Here, we show that multiple bulk or single-cell sequencing from the same patient contain recoverable information on these important quantities that can be recovered with evolutionary theory. This allows inferring in vivo cell mutation and cell survival rates in tissues of individual humans from single time point sequencing data.

We draw our inferences by defining and quantifying the distribution of mutational distances amongst multiple samples. We first discuss the required theoretical considerations and derive an analytical expression for the expected distribution of mutational distances from multi-sample sequencing data. We introduce a Bayesian sampling framework based on the mutational distance distribution, allowing us to disentangle mutation rates per cell division and cell survival/death rates. We apply this framework to whole-genome single-cell sequencing data of haematopoiesis and brain tissue and measure both evolutionary parameters during early development. Finally, we utilise multi-sample sequencing data on 16 tumours to infer patient specific evolutionary parameters in human cancers.

## Results

**The distribution of mutational distances**. All cells in a human tissue must have descended from a most recent common ancestor cell (MRCA) that existed briefly during early development. Similarly, all cells in a sample of a tissue must have descended from a (different) MRCA that was present in that tissue at an earlier time (Fig. 1a). Mutations found in all cells of the sample (clonal mutations) were present in this MRCA. If we take multiple samples of the same tissue, we can reconstruct the mutational profile (all mutations carried by a single cell) of multiple ancestral cells (Fig. 1a). Typically, these ancestral cells differ in their exact mutational profile between one another, because mutations inevitably accumulate differently in distinct lineages (Fig. 1b). We use the differences of the mutational profiles between ancestral cells to construct the distribution of mutational distances. We define a mutational distance as the number of mutations different between any two ancestral cells (Fig. 1c). In the language of set theory, if ancestral cell 1 carries a set of mutations $A$ and ancestral cell 2 carries a set of mutations $B$, then

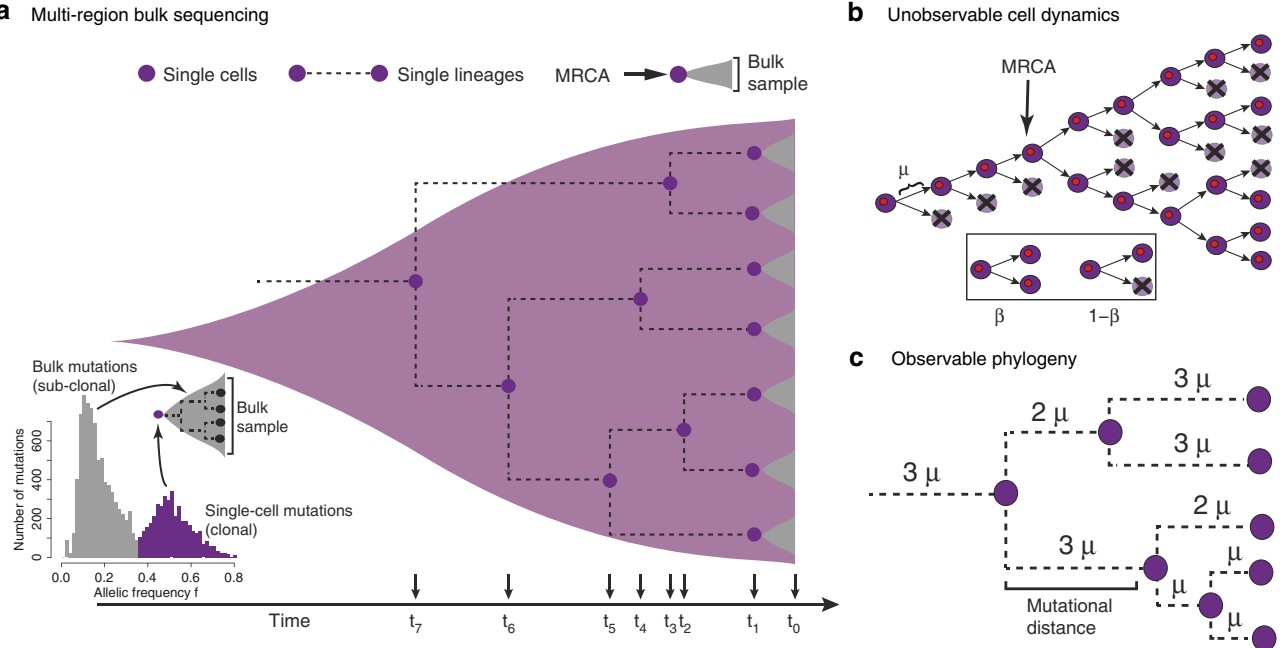

**Fig. 1 Multi-sample bulk sequencing encodes information on single cell lineages and single cell divisions. a** Each of the seven spatially separated tissue samples (in grey) consists of thousands to millions of cells that descended from a single most recent common ancestor (MRCA) cell. The genomic make-up of the single ancestral cell is described by the mutations clonal to the bulk sample. Those appear at high variant allele frequency in the sample (bottom-left panel, in purple). The intersection of mutations in any two bulk MRCA cells corresponds to the genomic profile of another more ancestral cell. This process continues back in time until the MRCA cell of all the sampled cells is reached. **b** The level of genomic variation within a growing tissue (e.g. development or cancer) is the direct consequence of mutation accumulation during cell divisions, leading to a branching structure. Importantly, the most fundamental parameters, the mutation rate $\mu$ and survival rate $\beta$ of cells per division that drive this process are not directly observable. **c** Mutation rate per division $\mu$ and cell survival rate $\beta$ leave identifiable fingerprints in the observable patterns of genetic heterogeneity within a tissue. Cell divisions occur in increments of natural numbers and thus the mutational distance between any two ancestral cells is a multiple of the mutation rate $\mu$.

by definition, both cells must have coalesced from an earlier ancestral cell (Fig. 1a). The mutational profile of this cell is given by the intersection $A \cap B$. This allows us to construct two mutational distances given by

$$y_1 = |A \setminus (A \cap B)| \quad \text{and} \quad y_2 = |B \setminus (A \cap B)| \qquad (1)$$

This process can be iterated for increasing combinations of samples per tumour.

We now turn to quantitative expressions for the expected distribution of mutational distances $P(y)$. In a single division, the probability of a cell to acquire $X$ novel mutations follows a Poisson distribution

$$P(X) = \frac{(\mu L)^X}{X!} e^{-\mu L}. \qquad (2)$$

Here, $\mu$ is the mutation rate (in units of base pairs per cell division) and $L$ the size of the sequenced genome. Throughout the paper, we assume a constant mutation rate and do not consider more punctuated catastrophic events or mutational bursts. Distances between cells of a lineage may arise from more than a single cell division. Instead, double, triple and higher modes of cell division contribute to the distribution of mutational distances of multiple samples. In general, a cell accumulates $X_1 + X_2 + \ldots + X_n$ number of novel mutations after $n$ divisions, which is again Poisson distributed.

In addition, we must account for cell death or differentiation, leading to lineage loss. We therefore introduce a probability $\beta$ of having two surviving lineages after a cell division and a probability $1 - \beta$ of a single surviving lineage (cell death). We can split the total of $n$ cell divisions into $r$ divisions that result in two surviving lineages (branching divisions) and $m$ divisions with only a single surviving lineage (non-branching divisions). The number of non-branching events $m$ is again a random variable, which follows a Negative Binomial distribution

$$P(m|r) = \binom{r + m - 1}{r - 1} \beta^r (1 - \beta)^m. \qquad (3)$$

The number of mutations acquired between two branching divisions depends jointly on the Poisson distributed number of mutations and the Negative binomial distributed number of non-branching divisions $m$. Formally, we can write for the total number of mutations between two branching divisions

$$Y = \sum_{i=1}^{m} X_i. \qquad (4)$$

Equation (4) is a random sum of random variables and different combinations of $X$ and $m$ imply the same mutational burden $Y$ within a single cell lineage. Intuitively, a measured mutational burden in a single lineage can result from either many non-branching divisions with a low mutation rate or, alternatively a few non-branching divisions with high mutation rate. The mutational burden of a single sample is insufficient to disentangle per-cell mutation and per-cell survival/death rates.

We therefore turn to the number of mutations different between ancestral cells. Suppose two ancestral cells are separated by $r$ branching divisions. Following from Eq. (4), we can calculate the probability distribution of the number of acquired mutations $P(y|r)$ after $r$ branching divisions

$$P(y|r) = \sum_{i=r}^{\infty} \binom{i - 1}{r - 1} \beta^r (1 - \beta)^{i-r} e^{-i\mu L} \frac{(i\mu L)^y}{y!}. \qquad (5)$$

Here the sum starts at $r$, as we need to have at least $r$ branching divisions and runs to infinity as in principal infinitely many non-branching divisions can occur (with vanishingly low probability). Finally, we need the expected distribution of branching divisions

$P(r)$ in a growing population of cells, which follow from coalescence theory[20–22]. For a growing population, e.g. human tissues during early development or cancer growth, we find

$$P(r) = \frac{\exp\left(-\frac{e^{-\beta(r+1)}}{\beta}\right) - \exp\left(-\frac{e^{-\beta r}}{\beta}\right)}{1 - \exp\left(-\frac{e^{-\beta}}{\beta}\right)}. \qquad (6)$$

We provide a more detailed derivation in the Methods. Combining Eqs. (5) and (6) we arrive at the final expression for the expected distribution of mutational distances in an exponentially growing population

$$P(y) = \sum_{r=1}^{\infty} \sum_{i=r}^{\infty} P(r) \binom{i - 1}{r - 1} \beta^r (1 - \beta)^{i-r} e^{-i\mu L} \frac{(i\mu L)^y}{y!}. \qquad (7)$$

The two evolutionary parameters of interest here, the mutation rate per cell division $\mu$ and the cell survival rate $\beta$, disentangle in Eq. (7). There are approximately four possible regimes for the distribution of mutational distances, discriminated by uni- or multimodality determined by combinations of small or large $\mu$ and $\beta$. In Fig. 2a we show four representative realisations of Eq. (7). The distribution of mutational distance is unimodal for sufficiently small mutation rate $\mu$ (bottom panels in Fig. 2a) with a single peak at the mean mutational distance $\mu L$. The per-cell survival probability $\beta$ determines the weight of the distribution towards larger distances. For $\beta = 1$ the distribution is sharply located around the mean mutation rate. However, for smaller $\beta$ more weight is given to larger distances and the distribution gets a fat tail. The same is true for the case of high mutation rate $\mu$, except the distribution is multi-modal with peaks separated by multiples of the mean mutational distance $\mu L$ (Fig. 2a). Again, $\beta$ determines the weight to higher mutational distances with lower $\beta$ causing a distribution with a long oscillating tail (top right panel in Fig. 2a). Note, the $y$-axes in Fig. 2a correspond to the probabilities of observing certain mutational distances. Lower probabilities require a higher resolution and therefore more sampling to resolve the exact shape of the distribution. In practice, the distribution of mutational distances is easiest to recover from data with low $\mu$ and high $\beta$ (fewest number of tissue samples required), whereas most samples are required for high $\mu$ and low $\beta$ (top right panel in Fig. 2a).

**Computational validation and MCMC inference framework.** We implemented stochastic spatial simulations of mutation accumulation in growing tissues using previously published code[23]. Briefly, cell birth and death on a two- or three-dimensional grid was simulated using a Gillespie algorithm[24]. During division, cells accumulate a number of new mutations drawn from a Poisson distribution. Simulations were stopped when the tissue reached ~1 million cells. This allowed us to take samples (either single cells or bulks) and construct all pairwise mutational distances of all ancestral cell lineages detectable in the samples. In Fig. 2b we show an example of the mutational distance distribution derived from 200 samples of a stochastic simulation (dots) compared to the theoretical prediction (dashed line).

We want to infer the microscopic evolutionary parameters $\mu$ and $\beta$ given a measured distribution of mutational distances. This can be done by Markov chain Monte Carlo methods (MCMC). We implemented a standard Metropolis-Hastings algorithm. In brief, a random pair of parameters $\mu$ and $\beta$ is drawn from uninformed uniform distributions and the likelihood of the model parameters given the data is calculated. The new set of parameters is accepted with a probability proportional to the likelihood ratio of the new and old parameter set (see Methods for more details). This framework recovers the true underlying

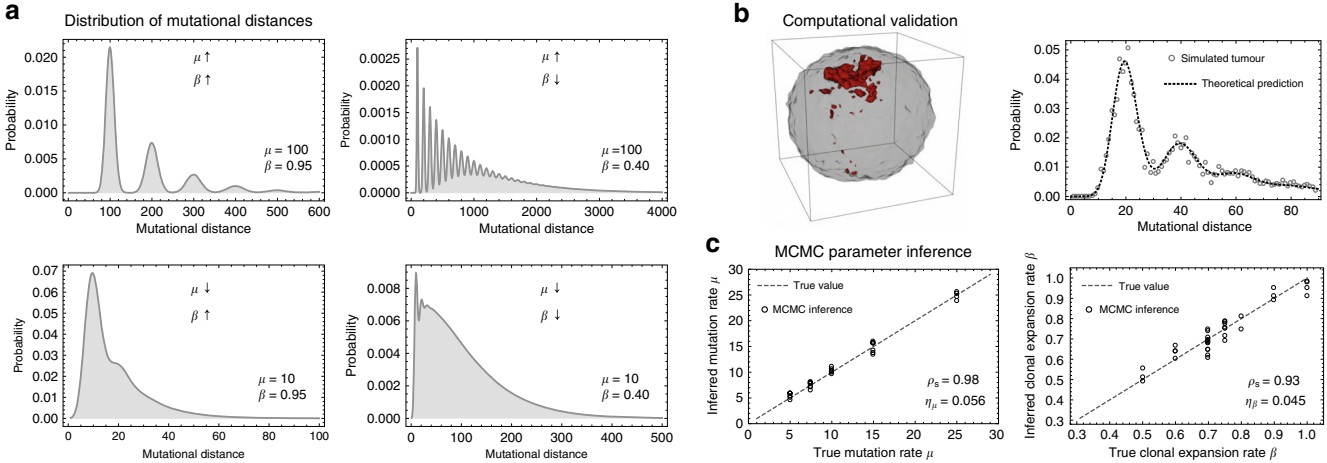

**Fig. 2 Distribution of mutational distances and computational validation. a** The quantised nature of cell divisions leads to a characteristic predicted distribution of mutational distances across cell lineages. The shape of the distribution depends on the exact values of $\mu$ and $\beta$. Roughly four different scenarios of combinations of small and large $\mu$ and $\beta$ are possible. They influence the shape of the distribution differently and thus constructing the distribution of mutational distances allows disentangling the mutation rate $\mu$ and cell survival rate $\beta$. **b** Spatial stochastic simulations confirm the ability of mutational distance distributions to disentangle mutation and lineage expansion rates (red area shows the spatial spread of a subclonal mutation). Dots show mutational distances inferred from 200 samples of a single stochastic computer simulation ($\mu = 20$, $\beta = 0.95$), the dashed line is the predicted distribution based on our Eq.(7). **c** A Monte Carlo Markov Chain inference framework based on mutational distance distributions reliably identifies mutation and lineage expansion rates in simulations of spatial and stochastically growing tissues (two-dimensional spatial stochastic simulations, $\mu$: Spearman Rho $= 0.98$, $p = 4 \times 10^{-23}$; $\beta$: Spearman Rho $= 0.93$, $p = 8 \times 10^{-16}$, Relative error: $\eta_\mu = 0.056$, $\eta_\beta = 0.045$).

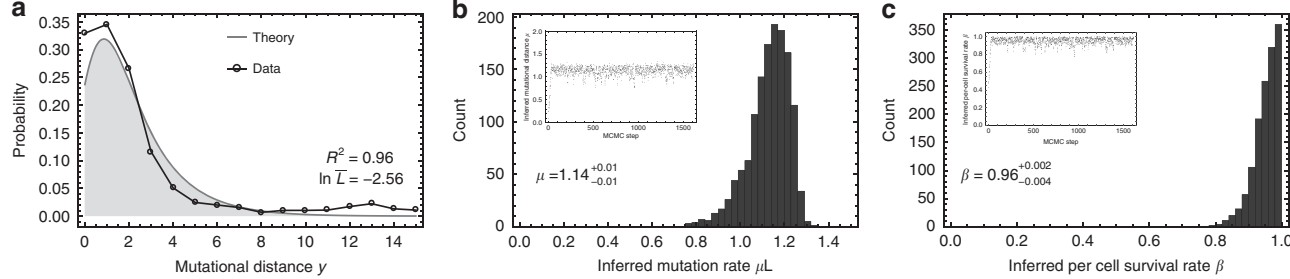

**Fig. 3 Per-cell mutation and per-cell survival rate inferences in healthy haematopoiesis during development. a** Mutational distance distribution inferred from 89 whole-genome sequenced healthy haematopoietic stem cells from ref. [13] (black dots), and best theoretical fit (grey line). Posterior parameter distribution of the MCMC inference for **b** the mutation rate per cell division ($\mu L = 1.14^{+0.12}_{-0.24}$ mutations per genome per cell division) and **c** the cell survival rate ($\beta = 0.96^{+0.038}_{-0.102}$). Median point estimates and 95% credibility intervals were taken from the posterior parameter distributions. The inferred mutation rate per cell division agrees with the original estimation of 1.2 mutations per cell division. Furthermore, our joined inference of mutation and cell survival rate confirms the original assumption of no cell death during early development of haematopoiesis.

parameters from stochastic simulations (Fig. 2c and Supplementary Figs. 17–21).

**In vivo mutation and cell survival rate inference in healthy haematopoiesis during early development.** We discuss the in vivo mutation accumulation in healthy haematopoiesis during early development as a first application. The cell population is growing and we expect a low mutation rate and a high per-cell survival rate during the development of early haematopoiesis[13,25]. In a recent study, Lee-Six and colleagues[13] sequenced the genome of 89 healthy haematopoietic stem cells of a single 59-year-old man and subsequently constructed the phylogeny of healthy haematopoiesis. They estimated the per-cell mutation rate to be 1.2 mutations per genome per division during early development assuming perfect cell doublings. Using the same data we construct the pairwise mutational distances of all ancestral cells limited to the 20 earliest branching events. The resulting distribution of mutational distances is shown in Fig. 3a. We then use the same MCMC framework discussed above to jointly infer the mutation and cell survival rate. The MCMC algorithm rapidly converges to

a fixed set of parameters (Supplementary Fig. 17). In Fig. 3a, b we show the posterior parameter distributions after an initial burn in phase of 200 MCMC steps. In agreement with Lee-Six and colleagues, we find a mutation rate of $\mu = 1.14^{+0.12}_{-0.24}$ mutations per genome per division (shown is the medium mutation rate per bp/cell-division and 95% credibility intervals inferred from the MCMC posterior parameter distribution), which corresponds to a mutation rate of $\mu = 3.9 \times 10^{-10}$ base pairs/division (assuming $3 \times 10^9$ bp in the human genome). Furthermore, we infer a per-cell survival rate of $\beta = 0.96^{+0.038}_{-0.102}$, independently confirming the original assumption of almost perfect cell doubling during early development[13].

**In vivo mutation and cell survival rate inference in single neurons during development.** In a recent publication, Bae et al.[12] collected single neurons from three fetuses 15 to 21 weeks post conception. Cells were expanded in culture and the whole-genome was sequenced. Here we focus on the case where 14 whole-genome sequenced single neurons were available (one fetus 17 weeks after conception). Again, we inferred all pairwise

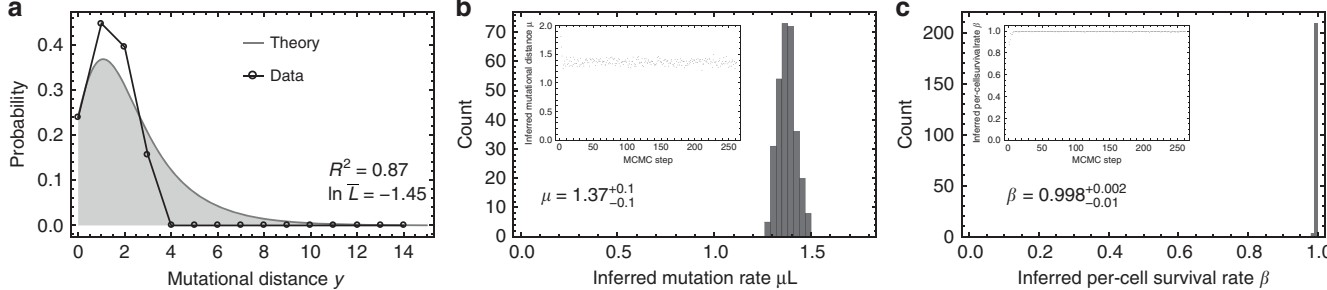

**Fig. 4 Per-cell mutation and per-cell survival rate inferences in single neurons during development. a** Mutational distance distribution inferred from 14 whole-genome sequenced single neurons from ref. [12] derived from one fetus (17 weeks past conception) (black dots), and best theoretical fit (grey line). MCMC inference for **b** the mutation rate per cell division ($\mu L = 1.37^{+0.1}_{-0.1}$ mutations per genome per cell division) and **c** the per-cell survival rate ($\beta = 0.998^{+0.002}_{-0.01}$). Median point estimates and 95% credibility intervals were taken from the posterior parameter distributions. The inferred mutation rate per cell division agrees with the original estimation of 1.3 mutations per cell division. Furthermore, our joined inference of mutation and cell survival rate confirms the original assumption of no cell death during early brain development.

mutational differences, constructed the corresponding distribution of mutational distances (Fig. 4a) and used our MCMC framework for joint parameter estimates. The MCMC converges rapidly and we find sharply localised posterior distributions for the mutation and cell survival rate. We infer a median mutation rate of $\mu = 1.37^{+0.1}_{-0.1}$ mutations per genome per division (corresponding to a mutation rate of $\mu = 4.6 \times 10^{-10}$ base pair/division) and a per-cell survival rate of $\beta = 0.998^{+0.002}_{-0.01}$. This inference agrees with Bae and colleagues original estimate of 1.3 mutations per genome per division based on a weighted average of all three fetuses, again assuming no cell death during early development. It also agrees with estimates of 1.2 mutations per division from de novo SNVs in familial trios[26]. The almost identical mutation rates in haematopoietic and brain tissue during early development may not be surprising. We would expect the DNA duplication and repair machinery to be stable across tissues during early development. It may even remain stable throughout live, as suggested by the linear rate of mutation accumulation with age across individuals[27–29].

**In vivo mutation and cell survival rates in human tumours.** We then investigated the per-cell mutation and survival rates in individual tumours. We analysed whole genome or exome sequencing of 131 biopsies from 16 tumours comprised of 1 colon adenoma, 7 colon carcinomas, 5 clear cell renal carcinomas and 2 lung squamous cell carcinomas (Table 1). When whole-genome sequencing was available, the mutational load was sufficient to apply the inference framework to each chromosome separately (Fig. 5 and Supplementary Figs. 1–9). The analysis was restricted to regions of chromosomes with same copy number profile in all samples of a tumour and inferences were normalised by copy-number and genome content. The resolution to infer the distribution of mutational distances from tumours was lower compared to healthy haematopoiesis or brain during development. Nevertheless, in most cases, the reconstructed distributions recover important features of the theoretical distribution (Supplementary Figs. 1–9 and 14). We found that mutation rates per cell division were 4–100 times higher in tumours compared to healthy tissue, ranging from $2.91 \times 10^{-9}$ (bp/division) in the colon adenoma to $53 \times 10^{-9}$ (bp/division) in one lung squamous cell carcinoma (Table 1). Mutation rates differ significantly between patients but not across chromosomes of the same patient (Supplementary Figs. 11 and 12). Overall this suggests important differences in mutation accumulation at the single cell level between tumours and is in agreement with recent experimental in vitro single cell mutation rate inferences[29,30].

**Table 1 Data summary and evolutionary parameter inferences.**

| Tissue type | Sequencing | # Samples | $\mu \times 10^{-9}$ | $\beta$ | Source |
|---|---|---|---|---|---|
| HSC (development) | Whole genome | 89 | 0.39 | 0.96 | Lee-Six |
| Neuron (development) | Whole genome | 14 | 0.46 | 0.99 | Bae |
| CRA | Exome | 6 | 2.91 | 0.46 | Cross |
| CRC (MSS) | Exome | 13 | 30.1 | 0.84 | Cross |
| CRC (MSS) | Exome | 8 | 12.5 | 0.43 | Cross |
| CRC (MSS) | Whole genome | 6 | 24.0 | 0.65 | Cross |
| CRC (MSS) | Whole genome | 7 | 10 | 0.51 | Cross |
| CRC (MSS) | Whole genome | 9 | 8.9 | 0.45 | Roerink |
| CRC (MSS) | Whole genome | 9 | 9.9 | 0.50 | Roerink |
| CRC (MSI) | Whole genome | 9 | 30.9 | 0.34 | Cross |
| CRC (MSI) | Whole genome | 7 | 17.9 | 0.47 | Roerink |
| CCRCC | Exome | 8 | 21.7 | 0.66 | Gerlinger |
| CCRCC | Exome | 11 | 31.2 | 0.86 | Gerlinger |
| CCRCC | Exome | 8 | 15.8 | 0.47 | Gerlinger |
| CCRCC | Exome | 8 | 2.3 | 0.80 | Gerlinger |
| CCRCC | Exome | 8 | 2.1 | 0.72 | Gerlinger |
| NSLC | Exome | 7 | 53 | 0.36 | Jamal-Hanjani |
| NACLC | Exome | 7 | 14 | 0.59 | Jamal-Hanjani |

The data of healthy tissue during development was taken from Lee-Six et al.[13] and Bae et al.[12]. Data on colorectal cancers is from Cross et al.[42] and Roerink et al.[43], data on renal cell carcinoma from Gerlinger et al.[44] and data on lung carcinomas from Jamal-Hanjani et al.[45]. Estimates for mutation and cell survival rates are from best MCMC fits based on the distribution of mutational distances.

To further unravel the underlying differences in mutation accumulation during tumour growth, we decomposed somatic mutations into the most prevalent trinucleotide mutational signatures[31] for three whole-genome sequenced colorectal carcinomas and inferred per-cell mutation and per-cell survival rates per signature in each chromosome (Fig. 5). Again, we find significant differences between patients (Supplementary Fig. 13), further supporting inter-tumour differences of mutation accumulation at the single cell level.

The inter-patient variation of the cell survival rate was evident. Whereas in healthy tissue almost all cells survive during development, in tumours cell survival rates vary between 0.34 in one MSI+ colon carcinoma and 0.86 in one renal cell carcinoma (Table 1). Again, per-cell survival rates were overall consistent if inferred from chromosomes of individuals, but varied significantly between patients (Fig. 6 and Supplementary Fig. 12). The underlying reasons for this inter-patient variation may be cell intrinsic and/or extrinsic, e.g. high cell death due to genomic instability, high mutational burden or immune surveillance. It will be of high interest to further unravel these differences on a patient specific basis in future studies. It should be noted that the inferred cell survival rates are high compared to previous estimates[32,33]. This is a direct consequence of the joint inference of mutation and cell survival rates that was not possible in earlier work.

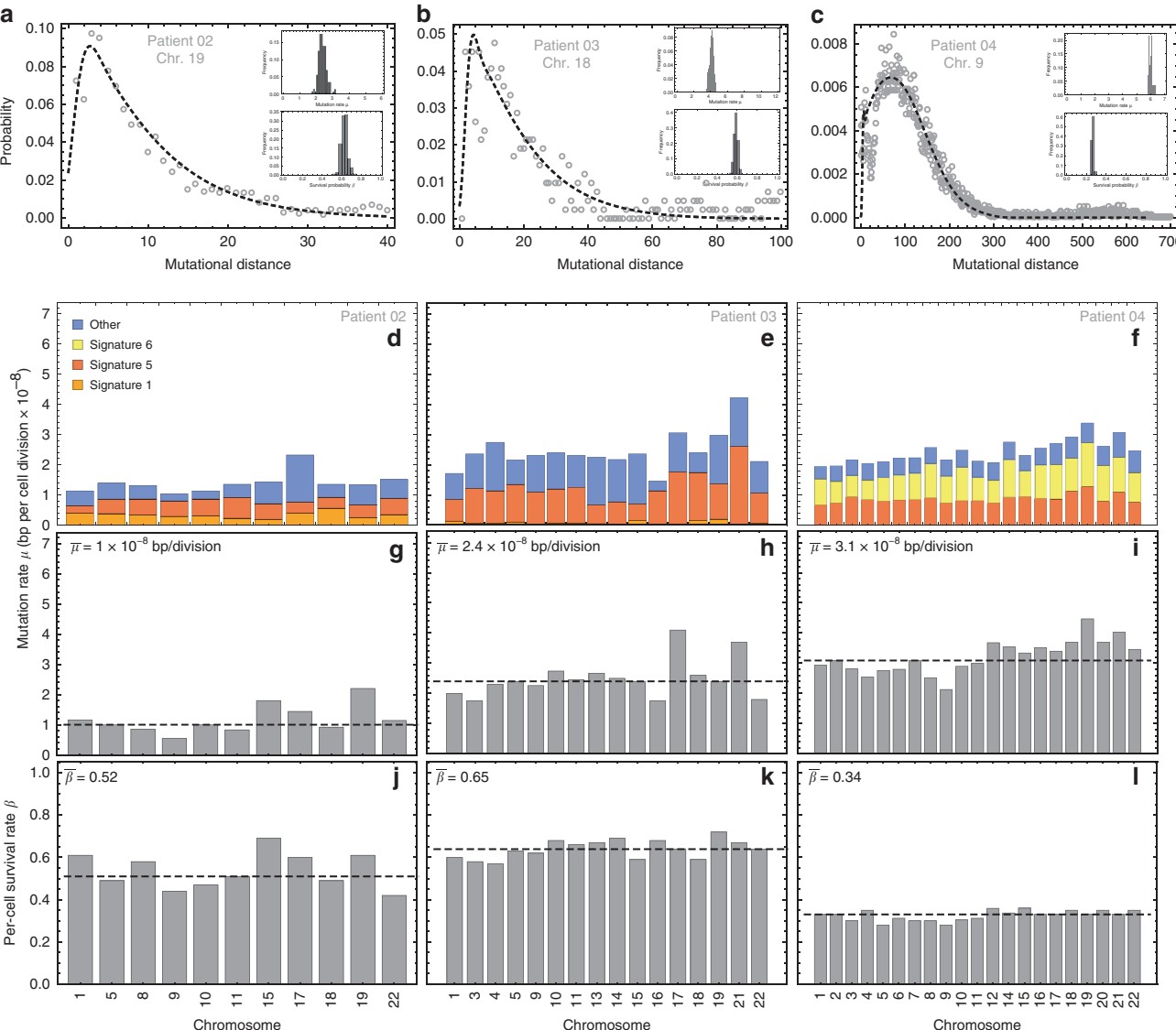

**Fig. 5 Mutational distance for three colorectal tumours. a–c** Examples of the mutational distance distribution on single chromosomes for three different colorectal carcinomas for which 6, 7 and 9 multi-region bulk samples were sequenced at whole-genome resolution (dots = data, dashed line = theoretical prediction based on MCMC parameter estimates—see insets). The distribution of mutational distances differs between patients, with Patient 04 (MSI Microsatellite Instability) showing one order of magnitude larger mutational distances. **d–f** Per-cell mutation rate per chromosome separated by trinucleotide mutational signature. Results are consistent across chromosomes, as expected (Methods). **g–i** The mean overall mutation rates are $(\mu_{02} = (1.0^{+0.46}_{-0.07}) \times 10^{-8}, \mu_{03} = (2.4^{+0.41}_{-0.19}) \times 10^{-8}$ and $\mu_{04} = (3.1^{+0.35}_{-0.12}) \times 10^{-8}$ bp/division, dashed lines), 20–60 times higher compared to healthy somatic cells. Patient 04 is MSI+ highlighted by signature 6. **j–l** Estimates of per-cell survival rates per chromosome are consistent across chromosomes of the same patient (Median: $\beta_{02} = 0.51^{+0.05}_{-0.05}, \beta_{03} = 0.65^{+0.02}_{-0.02}, \beta_{04} = 0.34^{+0.01}_{-0.01}$), but vary considerably between patients (Supplementary Fig. 12).

## Discussion

Here we presented a framework that allows disentangling the microscopic evolutionary forces of mutation and survival rates per cell division in humans from single time point measurements. Leveraging data on mutations in healthy haematopoiesis[13] and brain tissue[12], we found, in agreement with previous estimates, mutation rates of 1.14 and 1.37 mutations per whole-genome per cell division. Mutation rates were 4–100 times higher in cancers and showed considerable inter-patient variation.

The inference framework presented here relies on some assumptions. Mutation and cell survival rates are kept constant trough time and spatial location. We do not consider significant changes in cell fitness during growth and/or spatial resource constraints. These limitations are more important for tumour specific inferences and less relevant for healthy tissue. The exact

temporal and spatial change of the underlying microscopic evolutionary parameters over the lifetime of an individual tumour remains an open question. In some cases, there is evidence for singular catastrophic events[34] and mutational signatures may change between resection and relapse[35]. However, it will also be important to disentangle mutation and cell population dynamic processes in these cases. A more fine-grained sampling over space and time is needed to better access if and how evolutionary parameters change within tumours. Given the technological advances in single cell genomics[36,37], sequencing of potentially thousands of single cells would lead to significant information gain. This will allow probing potential changes of these evolutionary parameters over time.

Furthermore, we expect the inter-patient variation of per-cell mutation and survival rates to directly influence clinically

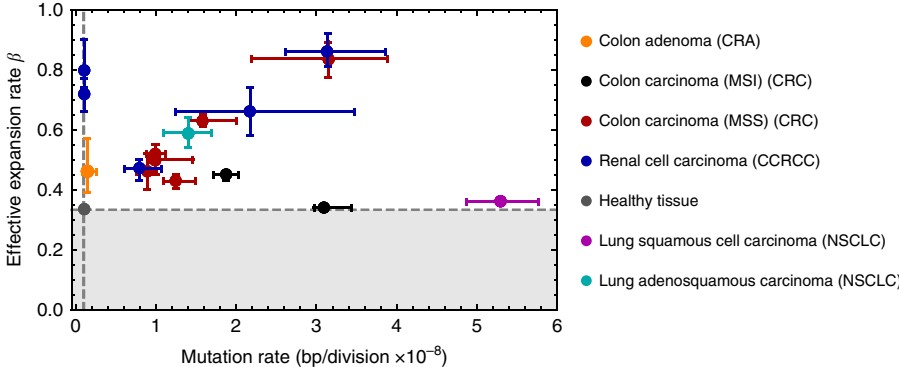

**Fig. 6 Map of per-cell mutation and per-cell survival rates across cancer types.** For each of the 16 tumours analysed we plot the per-cell mutation rate versus the per-cell survival rate. Median estimates and 95% credibility intervals for the mutation and cell survival rate are derived from the MCMC inferences as described in the main text. Dashed lines correspond to values of healthy tissue ($\mu_h = 1 \times 10^{-9}$, $\beta_h = 1/3$). White background corresponds to $\beta$ values that allow for growing cell populations as $\beta = 1/3$ corresponds to stable (homeostatic) populations. Shaded area describes values of $\beta$ that would lead to population extinction. Most cancers scatter across the map, indicating extensive inter-patient heterogeneity.

important variables, such as the likelihood of pre-existing treatment resistance[38], tumour age and aggressiveness[39]. Measuring microscopic evolutionary forces in humans allows for a mechanistic foundation for precision medicine.

## Methods

**Branching distribution in exponentially growing populations.** To calculate the expected distribution of branching events in an exponentially growing population, we can make use of coalescence theory[20,21]. Note that in coalescence theory one usually uses a backward time convention. If a population grows exponentially with $N(\tau) = e^{\beta\tau}$, coalescence considers backward time $t = -\tau$ such that populations effectively shrink exponentially. The probability of coalescence $P_\zeta(t)$ at time $t$ in an exponentially growing population is given by

$$P_\zeta(t) = \frac{1}{N(t)} \prod_{s=0}^{t-1} \left[1 - \frac{1}{N(s)}\right] \approx \frac{e^{\beta t}}{N_0} \exp\left(\frac{1 - e^{\beta t}}{\beta N_0}\right), \quad (8)$$

where $N(t)$ is the size of the growing population at time $t$. In our case, we are concerned with mutational distances and thus we ask for the distribution of times between coalescence events $\Delta t$ rather than the distribution of coalescence times $t$. However, we can directly infer this distribution from Eq. (8), by rewriting $\Delta t = t_0 - t$ as the time of the initiating cell population at some point in the past. By substituting $t_0 = \log(N_0)/(\beta)$, we have $\Delta t = \frac{\log(N_0)}{\beta} - t$ and we find for the distribution of times between coalescence events

$$P(\Delta t) = P\left(\frac{\log(N_0)}{\beta} - t\right) = e^{-\beta\Delta t} \exp\left(\frac{1 - N_0 e^{-\beta\Delta t}}{\beta N_0}\right). \quad (9)$$

This is for large $N_0$ well approximated by

$$P(\Delta t) = e^{-\beta\Delta t} \exp\left(-\frac{e^{-\beta\Delta t}}{\beta}\right). \quad (10)$$

We show the validity of this approximation in Supplementary Fig. 16. The normalized expression holds for all $N_0 \geq 1$. We can discretise this probability density function to derive at the probability for the number of branching divisions $r$ via

$$P(r) = \int_r^{r+1} d(\Delta t) P(\Delta t) = \int_r^{r+1} d(\Delta t) e^{-\beta\Delta t} \exp\left(-\frac{e^{-\beta\Delta t}}{\beta}\right)$$
$$= \exp\left(-\frac{e^{-\beta(r+1)}}{\beta}\right) - \exp\left(-\frac{e^{-\beta r}}{\beta}\right). \quad (11)$$

As we are interested in positive branch length only, we need to normalise the distribution for non-negative integers such that $1 = \frac{1}{C}\sum_{i=1}^{\infty} P(r = i)$. The normalising factor is $C = 1 - \exp\left(-\frac{e^{-\beta}}{\beta}\right)$, and the distribution of branching divisions $r$ in an exponentially expanding cell population becomes

$$P(r) = \frac{\exp\left(-\frac{e^{-\beta(r+1)}}{\beta}\right) - \exp\left(-\frac{e^{-\beta r}}{\beta}\right)}{1 - \exp\left(-\frac{e^{-\beta}}{\beta}\right)}. \quad (12)$$

Equation (12) together with Eq. (6) in the main text allows a complete description of the expected distribution of mutational distances in exponentially growing populations. It has to be noted that the coalescence approximation used here is based on a deterministic exponential growth function. It is known that such approaches do not always fully capture the full stochasticity especially at small population sizes and birth–death processes often perform better[22]. The individual based computer simulations used here are implementations of the Gillespie algorithm and are exact numerical representations of the underlying stochastic process. However, a further analysis on the stochasticity of the process for small population sizes is warranted.

**Interpretation of effective survival rate.** Throughout the paper we use the concept of the effective cell survival rate $\beta$. One can also formulate cell death with a microscopic perspective given a probability $\alpha$ for a daughter cell to die (or differentiate) after division. Such a probability allows for three outcomes after a cell division: with probability $(1 - \alpha)^2$ both daughter cells survive, with probability $2\alpha(1 - \alpha)$ one daughter cell survives and with probability $\alpha^2$ both daughter cells die. However, as we are bound to find surviving cell lineages in every possible measure of tumours, none of the observed cell lineages can have gone extinct. Mathematically, this implies that measurement conditions cell division on non-extinction of both daughter cells and we can write

$$\beta \equiv P(\text{successful division}|\text{non extinction}) = \frac{P(\text{successful division \& non extinction})}{P(\text{non extinction})}.$$

With the corresponding probabilities $\alpha$ we get

$$\beta = \frac{(1 - \alpha)^2}{1 - \alpha^2} = \frac{1 - \alpha}{1 + \alpha}. \quad (13)$$

We also can rearrange Eq. (13) to solve for $\alpha$,

$$\alpha = \frac{1 - \beta}{1 + \beta}. \quad (14)$$

If we interpret $\alpha$ as the probability of random cell death after a division, $\alpha$ must be smaller than 1/2. If $\alpha$ were larger than 1/2, tumour populations extinct almost surely after sufficiently many cell divisions. This implies $\beta > 1/3$ for growing populations.

**Simulations of mutation accumulation in growing tissues.** We simulated cell populations of ~1 million cells on a grid with varying birth–death and mutation rates using an implementation of the Gillespie algorithm based on code published in ref. [23]. The code is available at https://github.com/sottorivalab/CHESS.cpp. A cell division produces two surviving cells with probability $\beta$ or one surviving cell with probability $1-\beta$. During each division, each daughter cell inherits the mutations of its parent and in addition accumulates novel mutations. The number of novel mutations is drawn from a Poisson distribution with mean $\mu$. During simulations, the mutations for each cell as well as the division history of each cell are recorded.

We took samples (between 1 and 10k cells per sample) from each simulated tumour. For most inferences, we used maximal distance sampling. Sequencing errors were simulated for each bulk by binomial sampling assuming sequencing depths of 100x, by generating dispersed coverage values for input mutations. We do that by sampling a coverage from a Poisson distribution: Poisson $(\lambda = Z)$ with mean $\lambda$ equal to a desired sequencing depth $Z$. Once we have sampled a depth value $k$ for a mutation, we sample its frequency (number of reads with the variant allele frequency) with a Binomial trail. We use $f \sim \text{Binomial}(n, k)$, where $n$ is the proportion of cells carrying this mutation given all cells sampled in the simulated biopsy. This generates realistic mutation distributions comparable to available genomic sequencing data.

**Bayesian parameter inference**. We use a MCMC to recover the mutational distance $\mu L$ and the cell survival rate $\beta$ given a measured distribution of mutational distances. More precisely we implemented a standard Metropolis-Hastings-algorithm following below steps:

(i) Create a new random set of model parameters $w$ given the current set of parameters $v$ from a defined probability density $Q$, such that $Q(x|y) = Q(y|x)$.

(ii) Calculate the likelihood $L(P(w))$ of the model distribution $P(w)$ given the data.

(iii) Calculate the ratio of the new and old likelihood $\rho = L(P(w))/L(P(v))$. Accept the new parameter set with probability $\rho$ otherwise reject.

(iv) Repeat.

In our case the model distribution is given by Eq. (7) in the main text. To calculate the likelihood of Eq. (7) given the data, we have to choose a cut off for the infinite sums. However, real data always has a maximum mutational distance. Higher terms of the infinite sums contribute to higher mutational distances. The distribution of interest does not change for a sufficiently high cut off and each observed data set only requires finite many terms. Here we used $r = i = 30$ as upper cut-off, which is a conservative choice. We used uninformed uniform prior distributions for mutational distance $\mu L$ and the per-cell survival rate $\beta$ in all cases. Point estimates were extracted as sample medians from the MCMC inferences. The ranges of the uniform priors were adjusted to optimise acceptance rates and computational time. In our implementation, a new set of parameters is relative to the previously accepted parameter set $w_{New} = w_{old} + \Phi(w)$, where $\Phi$ is the prior parameter distribution. A typical range used in our inference scheme is $\Phi_{uniform}(\beta) = [-0.06, +0.06]$ and $\Phi_{uniform}(\mu) = [-0.15, +0.15]$. We also tested Gamma prior distributions and did not see differences in convergence. One numerical realisation of the Log-Likelihood function is shown in Supplementary Fig. 18 and example traces of the MCMC algorithm are shown in Supplementary Fig. 17. We also tested the influence of sequencing depth and spatial sampling strategies on the performance of the MCMC inference framework (Supplementary Figs. 19 and 20). The code for the MCMC inference is available at https://github.com/sottorivalab/MCMC-MutationalDistances-.

**Mutational signature analysis**. For each sample we found the set of signatures (among those signatures reported in CRC) that best explained the totality of mutations in the sample. We did a non-negative regression of the sample's mutations against all the CRC signatures[40] and found those signatures with non-zero coefficients. We took these as the candidate signatures for each sample.

For each mutation in each sample, we determined the likelihood of the mutation under each of the candidate signatures. We assigned a mutation to a candidate signature where the likelihood under that signature was at least twice that under any other. If there was no such signature, we assigned the mutation to "Other". The method was originally developed in[40] and is based on the R-package "SomaticSignatures"[41]. We did not adjust for differences in nucleotide composition when calculating differences between coding and non-coding regions as we wanted to infer the overall point mutation rate in these regions. Nucleotide dependent mutation rate estimates are shown in Supplementary Figs. 10 and 15. Nucleotide composition was adjusted for to calculate the mutation rates of mutational signatures using standard tools[41].

**Reporting summary**. Further information on research design is available in the Nature Research Reporting Summary linked to this article.

## Data availability

Sequencing data from healthy haematopoiesis is available from Lee-Six et al.[13], brain data during early development from Bae et al.[12], colorectal cancer data from Cross et al.[42] and Roerink et al.[43], renal cell carcinoma data from Gerlinger et al.[44]. and lung carcinoma data from Jamal-Hanjani et al.[45].

## Code availability

The code for stochastic simulations of tumour growth is available at https://github.com/sottorivalab/CHESS.cpp. The code for the MCMC inference is available at https://github.com/sottorivalab/MCMC-MutationalDistances-.

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

## Acknowledgements

A.S. is supported by the Wellcome Trust (202778/B/16/Z) and Cancer Research UK (A22909). T.G. is supported by the Wellcome Trust (202778/Z/16/Z) and Cancer Research UK (A19771). We acknowledge funding from the National Institute of Health (NCI U54 CA217376) to A.S and T.A.G. This work was also supported by Wellcome Trust funding to the Centre for Evolution and Cancer (105104/Z/14/Z). C.P.B. acknowledges funding from the Wellcome Trust (209409/Z/17/Z).

## Author contributions

B.W. and A.S. conceived the study. B.W. and J.C. performed mathematical analysis. B.W., M.J.W., K.C., D.T., J.F.M., G.D.C., D.N. W.C., I. S., W.H. & I.T. contributed to data analysis and simulations. A.S. and T.A.G. supervised the study. B.W., C.P.B., T.A.G. and A.S. wrote the paper. All authors read and approved the paper.

## Competing interests

The authors declare no competing interests.
