## [Peer Review File · Nature Communications]

Reviewers' comments:

Reviewer #1 (Remarks to the Author):

Introduction:

- Werner et al. present an interesting mathematical framework and approach to infer a point mutation rate per cell division (μ) and a cell death/differentiation probability per cell division (β) from somatic point mutation calls in multi-region sequencing experiments. They derive an analytical form for the expected distribution of mutational distances between samples under different values of μ and β , which they employ to infer the bayesian posterior parameter estimates from real datasets using an MCMC approach with the Metropolis-Hasting sampling.
- They apply this approach to 2 published datasets of embryonic development (one brain and one hematopoietic) and 16 tumours from 4 published datasets including cancer types with high mutation loads.
- They find that their approach fits data generated using a 3D tumour growth simulator; recapitulates independent estimates of μ and β in embryonic development datasets; and give interesting results in the tumour samples, consistent when inferred from independent mutation sets across chromosomes and different rates per mutational signatures. They also estimate tumour age from the cell survival rate.
- In summary, Werner et al. propose an interesting and novel approach to estimate healthy or tumour human tissue growth parameters, namely a point mutation rate per cell division and a cell death/differentiation rate per cell division from point mutations calls in multi-region sequencing data.

Main remarks:

- The code to perform the inference is not available.
- The authors develop an interesting probabilistic framework to disentangle per-cell-division mutation and cell death rates from point mutation calls from multi-region sequencing and apply it to several datasets. Like any probabilistic framework, it comes with assumptions. Given that this framework is the main result here, these assumptions should be better discussed:
 - 1) "the probability to acquire X novel mutations follows a Poisson distribution". It should be made clear that this is an assumption and the authors could contrast with other potential models. For example, what about the impact of punctuated events and bursts of mutations (e.g. kataegis)?

2) "the probability to acquire X novel mutations follows a Poisson distribution with constant rate along tumour growth": whereas some studies suggest that μ is similar in stem cells across tissues and different patient ages for given signatures, others show that clock-like mutations accumulate at different rates across tissues (literature by Alexandrov) and there seems to have an important acceleration of the clock in tumours when looking at relapse samples (doi: <http://dx.doi.org/10.1101/161562>).

3) "cell death/differentiation rates is constant over time": studies suggest that proliferation and thus likely cell death rates are not constant nor spatially-homogeneous during tumour growth (e.g. doi:10.1038/nature11344 and discussed in: doi:10.1371/journal.pcbi.1004731.g001), rather high cell death as to kick in later, as tumour would simply not grow otherwise.

These assumptions and their impact on the estimates should be discussed, as it might make it clearer that this framework is expected to work well on embryonic development datasets, but potentially much less so on tumour datasets.

- There is no information in the methods on how the sequencing data was analysed (mutation calls, copy number, etc.), and what was taken as input (e.g. BAMs vs. pre-analysed VCFs etc.). How do the authors deal with clonal vs. subclonal mutations; and low-purity samples, for which counts might be underestimated?

- The estimates of tumour age are not clearly explained and seem superfluous, as the authors suggest themselves to "interpret them with caution". The authors should either expand or remove this section.

Minor remarks:

- Line 614: "Nucleotide dependent mutation rate estimates are shown in SI Figure 6.": It is not clear in SI Figure 6 from the figure legend or caption where that is shown.

- Figure 4 d-f: it is not clear what is shown from the legend. If I understand each parameter was re-estimated individually per signature, which explains why the total does not match g-i.

- Supplementary Figure 11 shows that mutation rates vary between patient but not chromosomes of the same patients. Although a reasonable control, this is expected, as mutation load from different chromosomes is expected to be fairly stable. Perhaps another control would be to downsample the number of tumour samples per patient and get outputs from multiple runs there to compare the within vs. between patient variability.

- Regarding the MCMC results, it is difficult to explain why the parameters would be so different for different chromosomes (e.g. patient 2 chromosome 19, patient 4 chromosome 12). Is it due to very different mutation load on different chromosomes, or very different mutational distances? Some traces look like they do not evolve much from the start, it would be good to see the prior distributions plotted together with the posterior if that is possible.

Conclusions:

- Altogether, this is a widely interesting and novel story; though there is no code available, and the main text and the methods are missing important pieces of discussion and information.

Reviewer #2 (Remarks to the Author):

Werner and colleagues present a useful theory for predicting the mutational distance between samples from a supercritical branching process given the mutation rate and lineage survival rate. They validate this prediction against a previously published spatial simulation of stochastic tissue growth, sampling, and sequencing. Using MCMC, they then infer the mutation and survival rates from previously published human sequencing data in development (HSCs and neurons) and cancer (16 tumors of several types). In doing so, the paper not only outlines a potentially useful framework for understanding mutation accumulation in growing tissues, but also quantifies two parameters that are fundamental to these processes. Overall, this paper's findings are well-presented and will be of interest in multiple fields.

Major concerns:

The exponential-growth coalescent approximation from Eq. (19) is generally a poor approximation when population growth is modeled as a birth-death process rather than a deterministic expansion. One reason is because, when conditioned on survival, the expected growth of a tissue is faster than exponential (the "push of the past" phenomenon); another reason is that deterministic growth underestimates the variance in the coalescent time. Much better approximations are given by Stadler, Vaughan, Gavryushkin, et al. 2015, alongside the exact result (their Eq. 2.4). Would it be

possible to confirm that the four distributions plotted in your Fig. 1d would look similar even if you were to use these better approximations or the exact result?

Although the mutational distance between two cells descended from a common ancestor should include the mutations that accumulated along both lineages (assuming each mutation is unique), it seems that Y (Eqs. 6 and 18) as defined in this paper only counts the mutations that accumulated along one of the two lineages. Would you be able to clarify this point for me? To be specific, if two bulk samples are collected, then one sample has some MRCA anc1, and the other sample has some MRCA anc2 (which do not necessarily exist at the same time as depicted in Fig. 1a), and both samples then derive from their MRCA anc3, which in turn descends from the earliest cell in the tissue, anc4. Y seems to count the mutations that accumulated between anc4 and anc3 in addition to between anc3 and either anc1 or anc2. However, the mutational distance between the two samples should include the mutations that accumulated between anc3 and anc1 AND those which accumulated between anc3 and anc2, but not between anc4 and anc3. Can Eq. (6) be manipulated to address this issue? One idea would be to study the convolution of two iid copies of Y (i.e. Y_1+Y_2), but this would include two copies of the mutations that accumulated between anc4 and anc3, when those mutations would ideally not be included.

Why might the inferred survival rates for the 16 tumors in this study be so much greater than we typically see in the literature? An often-assumed value for the survival rate in cancer is $p=0.05$ (ranging from $p=0.01$ to $p=0.28$); see, for example, Bozic, Gerold, and Nowak 2016. Likewise, it seems that Chkhaidze, Heide, Werner, et al. 2019 used an initial death rate of 0.90 or 0.80 (survival rate of $p=0.10$ or $p=0.20$) in their Fig. S2. Even if this were converted to the survival rate as defined in this study, it would be $\beta = p^2/[2p(1-p)+p^2] = p/(2-p)$, which would range from $\beta=0.005$ to 0.163. To reflect the low survival rates in cancer, it would also be useful to run another batch of your grid-based tumor simulations with a death rate incorporated.

The discussion section is brief and could benefit from an overview of the limitations of the framework. As just one example, this framework assumes that the mutation and survival rates are constant over time and spatial location, even though these typically vary in cancer, particularly as driver mutations accumulate. As another example, the framework assumes that cell lineages are independent, and that there are no resource or spatial constraints.

Minor concerns:

The final sentence of the first paragraph of the results section (lines 94-96) defines mutational distance, a central concept in the paper, but it would be useful to rephrase this definition to ensure that it is as clear as possible.

The sections “The distribution of mutational distances” and “Distribution of mutational distances in multi-region sampling data” are almost identical, and there is no need to include both. I suggest removing the latter entirely. As a less ideal alternative, perhaps Eqs. (11-15) could be removed, since Eq. (16) follows directly from combining Eq. (8) with Eq. (9) using the basic idea of conditional probability. If the authors wish to provide the generating function for other reasons, then perhaps only Eq. (13) could be retained.

Eqs. (2) and (9) use the notation $P(m)$, but Eqs. (4) and (16) use the notation $P(Y_r = y)$, while Eq. (5) and (22-23) use the notation P_r . For consistency, it might be clearer to instead use $P(m|r)$, $P(y|r)$, and $P(r)$ respectively. Then Eqs. (6) and (18) would be simply $P(y)$.

Should Eq. (2) read $P(m|r) = \binom{r+m-1}{r-1} \beta^r (1-\beta)^m$ instead? Alternatively, it might be more useful to present the distribution of the total number of divisions n , $P(n|r) = \binom{n-1}{r-1} \beta^r (1-\beta)^{(n-r)}$, since it is n and not m that is needed later in Eq. (4).

Should line 152 introduce Eq. (3) as the total number of mutations that accumulate after r branching divisions, rather than just two branching divisions?

Fig. 1: Panels 1d and 1e are much too small to each contain four plots; can these be rearranged and enlarged? What do the red shaded regions in panel 1e-top-left represent? For panel 1e-top-right, what μ and β values were used, and were the samples maximally distant or randomly located? In line 215, no need to mention 2d grids unless you include those results.

Figs. 2-3: The x-axes for panels (a) and (b) are both labeled as μ , but they are different quantities; should they instead read (a) mutational distance y , and (b) inferred mutation rate $\mu \cdot L$? How was R^2 computed, and is there a better way to evaluate goodness of fit using the likelihood function?

Fig. 4: Could a brief explanation be provided as to why the total bar heights in panels d-f not correspond to the bar heights in panels g-i?

Table 1 and Fig. 5: It would be very useful to explain, preferably in the methods section, how tumor age was inferred from the survival rate, with references for the choice of 2-week (+/- 1 week) division times. It would also be useful to explain why $\beta_h = 1/3$ is the minimum feasible value (due to the 25:50:25 ratio of 0:1:2 surviving daughter cells).

In line 517, should the growth rate λ be set equal to one? In this same line, it might be beneficial to define N_0 . In line 512, it would be useful to clarify which population size $N(t)$ represents, and to clarify that time t is using the backwards-in-time convention.

In Eqs. (20-22), $\beta \cdot t$ should be replaced with $\beta \cdot (\Delta t)$, and dt should be replaced with $d(\Delta t)$.

In line 522, it would be useful to provide a more specific condition than “sufficiently large N_0 .” For example, $N_0 \gg \exp(\beta \cdot (\Delta t))$, or equivalently $\Delta t \ll t_0$, or equivalently $\beta \ll \log(N_0)/(\Delta t)$. For the HSC and neuron data, in which β is inferred to be nearly 1, is it still true that β satisfies this latter condition?

In line 566, there is a reference to SI Fig. 26b, but I cannot find this figure. I only seem to have SI Figs. 1-12. Am I missing some of your SI figures?

In line 588, do you mean $r = i = 30$ rather than $k = i = 30$? On this note, perhaps the index of summation i could be replaced by n for clarity in Eqs. (4), (6), and (14-18).

We want to thank both reviewers for the time and efforts invested to provide extremely valuable feedback on our submission. We were very happy to see that both reviewers find the manuscript interesting, novel and of importance to multiple fields. The comments helped us tremendously to improve the presentation and discussion of our results. Please find a detailed point by point response to all the comments and suggestions below.

Reviewer #1 (Remarks to the Author):

Introduction:

- Werner et al. present an interesting mathematical framework and approach to infer a point mutation rate per cell division (μ) and a cell death/differentiation probability per cell division (β) from somatic point mutation calls in multi-region sequencing experiments. They derive an analytical form for the expected distribution of mutational distances between samples under different values of μ and β , which they employ to infer the bayesian posterior parameter estimates from real datasets using an MCMC approach with the Metropolis-Hasting sampling.

- They apply this approach to 2 published datasets of embryonic development (one brain and one hematopoietic) and 16 tumours from 4 published datasets including cancer types with high mutation loads.

- They find that their approach fits data generated using a 3D tumour growth simulator; recapitulates independent estimates of μ and β in embryonic development datasets; and give interesting results in the tumour samples, consistent when inferred from independent mutation sets across chromosomes and different rates per mutational signatures. They also estimate tumour age from the cell survival rate.

- In summary, Werner et al. propose an interesting and novel approach to estimate healthy or tumour human tissue growth parameters, namely a point

mutation rate per cell division and a cell death/differentiation rate per cell division from point mutations calls in multi-region sequencing data.

We want to thank the reviewer again for the very positive comments on our work, which helped us to improve the manuscript significantly.

Main remarks:

- The code to perform the inference is not available.

We apologize for this oversight. The code for the stochastic computer simulations of tumour growth can be found at <https://github.com/kchkhaidze/CHESS.cpp>.

The code for the MCMC framework is now available on <https://github.com/BenWernerScripts/MCMC-MutationalDistances->

- The authors develop an interesting probabilistic framework to disentangle per-cell-division mutation and cell death rates from point mutation calls from multi-region sequencing and apply it to several datasets. Like any probabilistic framework, it comes with assumptions. Given that this framework is the main result here, these assumptions should be better discussed:

We agree with the reviewer. We now have considerably extended our discussion on the underlying assumptions for the probabilistic inference framework. We also included a discussion in which situations some of the assumptions might break down and some of the resulting consequences.

This discussion is now presented in Lines [143-145 & 408-424] in the revised manuscript.

1) "the probability to acquire X novel mutations follows a Poisson distribution". It should be made clear that this is an assumption and the authors could contrast with other potential models. For example, what about the impact of punctuated events and bursts of mutations (e.g. kataegis)?

We agree with the reviewer, this is certainly true. In our manuscript we are concerned with the clockwise process of mutation accumulation and how it is intertwined with different cell population dynamics in development and cancer. This process is unavoidable and universally present in somatic tissues. In addition, there might be certain singular catastrophic events of mutational bursts, chromosomal mis-segregations, chromosomal fusions amongst many others. This is beyond this work, but of great importance and now added as discussion to the manuscript.

This discussion is now presented in Lines [143-145 & 408-424] in the revised manuscript.

2) "the probability to acquire X novel mutations follows a Poisson distribution with constant rate along tumour growth": whereas some studies suggest that μ is similar in stem cells across tissues and different patient ages for given signatures, others show that clock-like mutations accumulate at different rates across tissues (literature by Alexandrov) and there seems to have an important acceleration of the clock in tumours when looking at relapse samples (doi: <http://dx.doi.org/10.1101/161562>).

We agree with the reviewer and also consider this an important issue. We first should distinguish healthy tissue and cancer. In healthy tissue there is increasing evidence for a stable mutation rate across ages and individuals (in many healthy tissues one finds linear dependences of tissue age and mutational burden across individuals of different ages). The situation is probably different in cancer.

First, we would expect that mutation rates differ significantly between tumours. We see evidence in our work here. Second, it is conceivable that mutation rates increase during tumour progression. We consider this a currently open question.

Studies based on mutational signatures are not yet able to disentangle mutation rate and cell population dynamics. We will need higher sampling density or higher resolved time series data to unravel the change of the mutation rate and cell population dynamics during tumour growth. This is a very important topic for future studies and is now discussed in more detail in the manuscript.

This discussion is now presented in Lines [143-145 & 408-424] in the revised manuscript.

3) "cell death/differentiation rates is constant over time": studies suggest that proliferation and thus likely cell death rates are not constant nor spatially-homogeneous during tumour growth (e.g. doi:10.1038/nature11344 and discussed in: doi:10.1371/journal.pcbi.1004731.g001), rather high cell death as to kick in later, as tumour would simply not grow otherwise.

These assumptions and their impact on the estimates should be discussed, as it might make it clearer that this framework is expected to work well on embryonic development datasets, but potentially much less so on tumour datasets.

We thank the reviewer for the comment and agree. A constant cell death/differentiation rate is an important assumption of the model. We now discuss it in more detail in the manuscript.

This discussion is now presented in Lines [143-145 & 408-424] in the revised manuscript.

- There is no information in the methods on how the sequencing data was analysed (mutation calls, copy number, etc.), and what was taken as input (e.g. BAMs vs. pre-analysed VCFs etc.). How do the authors deal with clonal vs. subclonal mutations; and low-purity samples, for which counts might be underestimated?

We apologize for a lack of detail on this point. All data presented in this manuscript has been published and is available for public use. We have used the

mutational calls as used in the original publications. We now added explanation to the supplement and refer to it in the main text. We also added an analysis how sequencing depth changes the inferences presented here. One advantage of our method is the reliance on clonal mutations in bulk samples (bulk specific sub-clonal mutations do not contribute to the inference as we do pairwise comparisons of joined mutations between bulk samples). Inferences remain stable for relatively low coverage (Figure R1). The analysis is now discussed and added to the supplement.

This discussion is now presented in Lines [580-588] and SI Figure 19 in the revised manuscript.

Figure R1: Parameter inference for different sequencing depth. Shown are the parameter inferences of the mutation rate **(a)** and the survival rate **(b)** for 10 spatial tumour simulations with $\mu = 15$ and $\beta = 0.8$ from the mutational distance distribution derived from 9 bulk samples with simulated sequencing depth of 25x, 50x and 200x. Shown are also the relative errors η for each scenario. The construction of the mutational distance distribution relies on the identification of clonal mutations within bulk samples. Consequently, the inferences remain accurate for a simulated sequencing depth of 25x.

- The estimates of tumour age are not clearly explained and seem superfluous, as the authors suggest themselves to "interpret them with caution". The authors should either expand or remove this section.

We agree with the reviewer. This inference was based on quite strong assumptions that probably differ between patients again. This inference is not essential to our manuscript and we have decided to remove it.

Minor remarks:

- Line 614: "Nucleotide dependent mutation rate estimates are shown in SI Figure 6.": It is not clear in SI Figure 6 from the figure legend or caption where that is shown.

We apologize. This was a formatting mistake. We added the corresponding Figure to the supplement and referenced it accordingly.

This is now SI Figure 10 in the revised manuscript.

Figure R2: Mutation rates of mutational subtypes. The mutation rate for mutational subtypes was inferred based on our MCMC algorithm for individual chromosomes (see Figure 5 in the main text) for all 3 patients separately and normalised for the C & T content at each chromosome. In Patient 02 and 04 transitions show higher mutation rates than transversions. Interestingly, in Patient 02 transversions T → A and T → G are absent, whereas in Patient 04 they are detectable. Patient 03 shows a distinct pattern of mutation accumulation. Here transitions and transversions appear equally likely, with C → X mutations slightly more likely compared to T → X mutations.

- Figure 4 d-f: it is not clear what is shown from the legend. If I understand each parameter was re-estimated individually per signature, which explains why the total does not match g-i.

The reviewer is correct. We extended the corresponding explanation in the caption of Figure 4 [Lines 318-319].

- Supplementary Figure 11 shows that mutation rates vary between patient but not chromosomes of the same patients. Although a reasonable control, this is expected, as mutation load from different chromosomes is expected to be fairly stable. Perhaps another control would be to downsample the number of tumour samples per patient and get outputs from multiple runs there to compare the within vs. between patient variability.

We agree with the reviewers suggestion. We now added additional tests. We now downsample (i) the data for healthy haematopoiesis and show (ii) downsampled data of patient 04 (in this case 9 independent samples were available and thus downsampling was possible). In addition we show that parameter inferences remain consistent if inferred on different parts of the genome.

This is now presented in SI Figures 14 & 15 in the revised manuscript.

Figure R3: Data down-sampling and parameter inferences. a),b) Parameter inferences for the down-sampled data of healthy haematopoiesis. **c),d)** Parameter inferences for the down-sampled data of chromosome 1 of patient 04.

Figure R4: Inference of per-cell mutation and per-cell survival rate for whole genome (per chromosome, open grey circles), non-coding (black squares) and coding mutations (red circles) in Patients 02-04. The coding mutation rate in patient 02 is slightly increased compared to whole genome inferences ($\mu_{WG}^{02} = 1 \times 10^{-8}$, $\mu_{Ex}^{02} = 2.8 \times 10^{-8}$), they are slightly lower in patient 03 ($\mu_{WG}^{03} = 2.4 \times 10^{-8}$, $\mu_{Ex}^{03} = 2.02 \times 10^{-8}$) and the same in patient 04 ($\mu_{WG}^{04} = 3.1 \times 10^{-8}$, $\mu_{Ex}^{04} = 3.08 \times 10^{-8}$). Non-coding mutation rates agree with median whole genome mutation rates.

- Regarding the MCMC results, it is difficult to explain why the parameters would be so different for different chromosomes (e.g. patient 2 chromosome 19, patient 4 chromosome 12). Is it due to very different mutation load on different chromosomes, or very different mutational distances? Some traces look like they do not evolve much from the start, it would be good to see the prior distributions plotted together with the posterior if that is possible.

We agree with the reviewer. Also, the most inferences are consistent, there are a few outliers. In case of chromosome 19 of Patient 02 and chromosome 12 of Patient 04 the mutational distances are larger (relative to the size of the chromosomes). At this point we do not know if this is caused by some differences

in biology (e.g. locally increased mutation rates) or potentially a genome doubling remained undetected for both chromosomes (which would appear as if the mutation rate of that chromosome doubled).

We also agree with the reviewer that some traces of the MCMC inferences seem very stable. In many cases we optimized the initial value of the MCMC trace to reduce computational time. The trace starts near optimum. In some cases, especially when we investigate whole genome sequencing of MSI cancers, calculating the likelihood function of the MCMC algorithm repeatedly is computationally costly and therefore optimizing the initial condition as well as length of the trace can improve performance considerably. In Figure R5 we show a few examples of the MCMC trace with different initial values and its convergence to a stable pair of parameter inferences.

Our MCMC framework is an implementation of the Metropolis-Hastings-algorithm. As such we have a proposal distribution that varies each parameter around its current value and accepts a new set of parameters proportional to the corresponding likelihood ratios of the old and new parameter set (we describe the algorithm in more detail in the supplement). One can use different distributions for the proposal distributions. In our implementation, we use a normal distribution, but also tested Gamma distributed proposal distributions, with no difference of the actual inferred parameter set.

We have extended our discussion in the manuscript now [Lines 571-577] and added SI Figures 17 & 18 to the revised manuscript.

Figure R5: Examples of the MCMC parameter estimation. Shown are multiple realisations of the MCMC algorithm for **a) & b)** healthy haematopoiesis (see also Figure ?? in the main text) and **c) & d)** Chromosome 19 of Patient 02 as shown in panel a) of Figure ?? in the main text. Each inference started with different initial conditions μ_0 and β_0 . We use a realisation of the Metropolis-Hastings algorithm and chains are converging to stable parameter pairs.

Conclusions:

- Altogether, this is a widely interesting and novel story; though there is no code available, and the main text and the methods are missing important pieces of discussion and information.

We want to thank the reviewer again.

Reviewer #2 (Remarks to the Author):

Werner and colleagues present a useful theory for predicting the mutational distance between samples from a supercritical branching process given the mutation rate and lineage survival rate. They validate this prediction against a previously published spatial simulation of stochastic tissue growth, sampling, and sequencing. Using MCMC, they then infer the mutation and survival rates from previously published human sequencing data in development (HSCs and neurons) and cancer (16 tumors of several types). In doing so, the paper not only outlines a potentially useful framework for understanding mutation accumulation in growing tissues, but also quantifies two parameters that are fundamental to these processes. Overall, this paper's findings are well-presented and will be of interest in multiple fields.

We want to thank the reviewer again for the very constructive and encouraging comments that helped us to improve the manuscript considerably. We were very happy to read that the reviewer considers the manuscript of interest to multiple fields.

Major concerns:

The exponential-growth coalescent approximation from Eq. (19) is generally a poor approximation when population growth is modeled as a birth-death process rather than a deterministic expansion. One reason is because, when conditioned on survival, the expected growth of a tissue is faster than exponential (the “push of the past” phenomenon); another reason is that deterministic growth underestimates the variance in the coalescent time. Much better approximations are given by Stadler, Vaughan, Gavryushkin, et al. 2015, alongside the exact result (their Eq. 2.4). Would it be possible to confirm that the four distributions plotted in your Fig. 1d would look similar even if you were to use these better approximations or the exact result?

We thank the reviewer for pointing us to this extremely interesting discussion of the potential limitations of the deterministic growth approximation of the coalescence approach. This is a fascinating read we were not aware of before.

We absolutely agree that this is an important issue. In our manuscript, we test the inference scheme on individual- based stochastic computer simulations of accumulating mutations in a growing tumour. Each possible reaction in the simulation (e.g. birth, death, mutation, spatial position etc.) are implemented by a Gillespie algorithm and thus are an “exact” numerical realisation of the underlying Master Equation of the stochastic process. Thus, the resulting mutational distance distributions derived from these simulations can be considered exact under the bounds of the imposed stochastic process.

We have now added additional tests of the inference scheme (in addition to the original Figure in the main manuscript) and the stochastic spatial simulations of tumour growth with a wider range of model parameters (Figures R??) and find good agreements.

As stated in manuscript by Tanja Stadler and colleagues: the deterministic approximation “is fine in case the number of generations where the population size is ‘very small’ is considered to be not ‘many’ “. This is probably what happens in our case. We feel that a complete resolution of this complex issue is beyond the scope of this manuscript. We now added this important point to the manuscript.

We have included discussion on this point in Lines [483-489] in the revised manuscript.

Although the mutational distance between two cells descended from a common ancestor should include the mutations that accumulated along both lineages (assuming each mutation is unique), it seems that Y (Eqs. 6 and 18) as defined in this paper only counts the mutations that accumulated along one of the two lineages. Would you be able to clarify this point for me? To be specific, if two

bulk samples are collected, then one sample has some MRCA anc1, and the other sample has some MRCA anc2 (which do not necessarily exist at the same time as depicted in Fig. 1a), and both samples then derive from their MRCA anc3, which in turn descends from the earliest cell in the tissue, anc4. Y seems to count the mutations that accumulated between anc4 and anc3 in addition to between anc3 and either anc1 or anc2. However, the mutational distance between the two samples should include the mutations that accumulated between anc3 and anc1 AND those which accumulated between anc3 and anc2, but not between anc4 and anc3. Can Eq. (6) be manipulated to address this issue? One idea would be to study the convolution of two iid copies of Y (i.e. Y_1+Y_2), but this would include two copies of the mutations that accumulated between anc4 and anc3, when those mutations would ideally not be included.

This is an interesting observation and we thank the reviewer for this remark. We agree with the reviewer that equations 6 & 18 appear to represent mutations along one lineage. It is important to remember that equations 6 & 18 do not count lineages but represent expected lineage length distributions (which as we keep the underlying model parameter constant must be symmetric for both lineages). As such they are independent of the exact number of lineages (given a sufficiently large sample size). Once the expected distribution of mutational distances is normalised, the dependence on lineage counts vanishes. For the purpose here, we always compare normalised mutational distance distributions.

Why might the inferred survival rates for the 16 tumors in this study be so much greater than we typically see in the literature? An often-assumed value for the survival rate in cancer is $p=0.05$ (ranging from $p=0.01$ to $p=0.28$); see, for example, Bozic, Gerold, and Nowak 2016. Likewise, it seems that Chkhaidze, Heide, Werner, et al. 2019 used an initial death rate of 0.90 or 0.80 (survival rate of $p=0.10$ or $p=0.20$) in their Fig. S2. Even if this were converted to the survival rate as defined in this study, it would be $\beta = p^2/[2p(1-p)+p^2] = p/(2-p)$, which would range from $\beta=0.005$ to 0.163. To reflect the low survival rates in cancer, it would also be useful to run another batch of your grid-based tumor simulations with a death rate incorporated.

We thank the reviewer for this remark. This is an important observation. In the original work of Bozic and colleagues it was not possible to disentangle mutation and survival rates. They therefore made the for the time (very reasonable) choice to assume a mutation rate of 5×10^{-10} per bp per division (typical germline mutation rate). The authors then extrapolated how many generations would be necessary to reach a certain typically observed driver and passenger mutational load. Together with assumed cell proliferation rates this leads to very low cell survival rate estimates.

Sequencing data of the last years shows that somatic mutations rates are likely higher in healthy tissues and probably much higher in many tumours. Bozic et al. discuss in their paper that results would be consistent with a death rate of 0 ($\beta = 1$ in our notation) if the mutation rate were 2×10^{-7} per bp per division (page 8 in Bozic et al. PLoS CB 2016). We find mutation rates of the order of 10^{-8} in many tumours which consistently would lead to much lower death rates in the work of Ivana Bozic and colleagues. We have added this to the discussions of the paper.

In our previous work of Chkhaidze et al. the parameter choice was arbitrary in a sense that we wanted to show different possible scenarios, but there is no direct inference of the model parameters on tumour specific data.

We discuss this now in the revised manuscript [Lines 394-397].

We also agree with the second point of the reviewer and have added another set of simulations with varying death rates to show that the inference framework also works for high death rates. This has been added as a supplemental figure and is accordingly referenced in the manuscript.

This additional analysis is now presented in SI Figures 19-21.

Figure R6: Spatial stochastic simulation inferences with varying per-cell survival rates. Panels (a)-(c) show examples for the mutational distance distribution reconstructed for cases of high mutation rate and different per-cell survival rates. The distributions are plotted with same y-axes to show the dramatic differences in the shape of the distributions (notice the different scales of the x-axis in panels a) to c)). The inset of panel (a) shows the same distribution, just with a differently scaled y-axis. Panels (d) & (e) show the inference of the evolutionary parameters for independent stochastic runs of spatial tumour simulations (9 bulk samples per simulation). Inferences are robust for low and high death as shown by relative errors η .

The discussion section is brief and could benefit from an overview of the limitations of the framework. As just one example, this framework assumes that the mutation and survival rates are constant over time and spatial location, even though these typically vary in cancer, particularly as driver mutations accumulate. As another example, the framework assumes that cell lineages are independent, and that there are no resource or spatial constraints.

We agree with the reviewer. We now have considerably extended our discussion and give a detailed account of the underlying model assumptions as well as limitations of the current framework.

We added an extended discussion to the revised manuscript [Lines 408-423].

Minor concerns:

The final sentence of the first paragraph of the results section (lines 94-96) defines mutational distance, a central concept in the paper, but it would be useful to rephrase this definition to ensure that it is as clear as possible.

We agree with the reviewer on this important point and have changed our explanation to the formulation:

“We define a mutational distance as the number of mutations different between any two ancestral cells, Figure 1c. In the language of set theory, if ancestral cell 1 carries a set of mutations A and ancestral cell 2 carries a set of mutations B , then the mutational distance y is the number of elements of the symmetric difference of A and B , $y = |A \ominus B|$.” [Lines 114-119] in the revised manuscript.

The sections “The distribution of mutational distances” and “Distribution of mutational distances in multi-region sampling data” are almost identical, and there is no need to include both. I suggest removing the latter entirely. As a less ideal alternative, perhaps Eqs. (11-15) could be removed, since Eq. (16) follows directly from combining Eq. (8) with Eq. (9) using the basic idea of conditional probability. If the authors wish to provide the generating function for other reasons, then perhaps only Eq. (13) could be retained.

We agree with the reviewer. The first paragraph in the supplement was repetitive and we have removed it as suggested.

Eqs. (2) and (9) use the notation $P(m)$, but Eqs. (4) and (16) use the notation $P(Y_r = y)$, while Eq. (5) and (22-23) use the notation P_r . For consistency, it might be clearer to instead use $P(m|r)$, $P(y|r)$, and $P(r)$ respectively. Then Eqs. (6) and (18) would be simply $P(y)$.

We thank the reviewer for this suggestion. It makes the notation more transparent and we happily adopt it to our manuscript.

Should Eq. (2) read $P(m|r) = \binom{r+m-1}{r-1} \beta^r (1-\beta)^m$ instead? Alternatively, it might be more useful to present the distribution of the total number of divisions n , $P(n|r) = \binom{n-1}{r-1} \beta^r (1-\beta)^{(n-r)}$, since it is n and not m that is needed later in Eq. (4).

Thanks for spotting this mistake. This is corrected now.

Should line 152 introduce Eq. (3) as the total number of mutations that accumulate after r branching divisions, rather than just two branching divisions?

Our idea was to first discuss mutation accumulation across a single lineage and show that a combination of different parameters can lead to the same mutational burden. This might be obvious from a theoretical perspective, but is often not really appreciated in cancer genomic analysis.

Fig. 1: Panels 1d and 1e are much too small to each contain four plots; can these be rearranged and enlarged? What do the red shaded regions in panel 1e-top-left represent? For panel 1e-top-right, what μ and β values were used, and were the samples maximally distant or randomly located? In line 215, no need to mention 2d grids unless you include those results.

We thank the reviewer and agree with his advice. We now split the original Figure 1 into two independent Figures, one showing the general framework and a second showing the computational inference.

The shaded regions show the spatial spread of a random subclonal mutation that occurred after the tumour was initiated.

The values to generate the mutational distance distribution in the previous panel 1e are $\mu = 20$ and $\beta = 0.95$. Spatial samples on the tumour were randomly distributed. However, based on simulations we find that for a low number of samples per tumour maximal distance sampling is slightly better than randomly

distributed samples. The information is now added to the caption of the figure and below analysis is added to the supplement.

The new Figure is now presented as SI Figure 20 in the revised manuscript.

Figure R7.: Random and Maximal Distance sampling. In order to test differences of random and maximal distance sampling, we did 10 spatial simulations with same underlying parameters (dashed lines). We then took 9 bulk samples either randomly or with maximal spatial distance and used our MCMC for parameter inferences. A maximal distance sampling strategy performs slightly better compared to random sampling (indicated by the relative errors η).

Figs. 2-3: The x-axes for panels (a) and (b) are both labeled as μ , but they are different quantities; should they instead read (a) mutational distance y , and (b) inferred mutation rate $\mu \cdot L$? How was R^2 computed, and is there a better way to evaluate goodness of fit using the likelihood function?

We thank the reviewer for noticing our labelling error. This has been corrected now. The R^2 was calculated as mean square distance of each binned mutational distance measured from the data to the best fit of the mutational distance distribution given the inferred pair of parameters. Below we also now show an example trace of the log likelihood of the MCMC fit of the healthy haematopoiesis data (Figure8). a) shows the entire trace, whereas in b) we show the accepted values of the MCMC realisation for the first 1000 MCMC steps.

Figure R8.:Example of Log Likelihood trace of the MCMC parameter estimation. Shown is the trace of the Log Likelihood function for a) the first 1000 steps of the MCMC algorithm for the parameter inference of healthy haematopoiesis. b) Accepted values of the MCMC for the first 1000 steps. After a burn-in phase of 100 steps, we find $\log L = -2.56$.

The new Figure is now presented (SI Figure 18 in the revised manuscript).

Fig. 4: Could a brief explanation be provided as to why the total bar heights in panels d-f not correspond to the bar heights in panels g-i?

We assigned mutational signature for each chromosome and then run the MCMC for each mutational signature and each chromosome independently. We added an additional explanation to the caption of Figure 4.

Table 1 and Fig. 5: It would be very useful to explain, preferably in the methods section, how tumor age was inferred from the survival rate, with references for the choice of 2-week (+/- 1 week) division times. It would also be useful to explain why $\beta_h = 1/3$ is the minimum feasible value (due to the 25:50:25 ratio of 0:1:2 surviving daughter cells).

We decided to take out the tumour age inferences as they were based on general assumptions that probably do not hold true on an individual patient basis.

We added an extended discussion why $\beta=1/3$ is the expected lower threshold for healthy haematopoiesis. This is now presented in Lines [492-518] in the revised manuscript.

In line 517, should the growth rate λ be set equal to one? In this same line, it might be beneficial to define N_0 . In line 512, it would be useful to clarify which population size $N(t)$ represents, and to clarify that time t is using the backwards-in-time convention.

Yes indeed, here we measure time in generations and λ should be set to 1. We also added discussion on the coalescence backward time convention to the supplement.

[Lines 446-449] in the revised manuscript.

In Eqs. (20-22), $\beta \cdot t$ should be replaced with $\beta \cdot (\Delta t)$, and dt should be replaced with $d(\Delta t)$.

We agree and have changed the equations accordingly.

In line 522, it would be useful to provide a more specific condition than “sufficiently large N_0 .” For example, $N_0 \gg \exp(\beta \cdot (\Delta t))$, or equivalently $\Delta t \ll t_0$, or equivalently $\beta \ll \log(N_0)/(\Delta t)$. For the HSC and neuron data, in which β is inferred to be nearly 1, is it still true that β satisfies this latter condition?

We agree with the reviewer. In our original manuscript, we were too vague in what N_0 sufficiently large means. Below we plot realisations of equations S2 and S3 for $\beta = 1$ and different values of N_0 . If we just plot equations S2 and S3, the approximation seems to fail for $N_0 = 1$. However, it is important to note that equations S2 and S3 are not normalized. In panel b) we show the normalized expressions. Once normalized both expressions are numerically identical. What is used for inference finally are the normalized expressions and therefore the approximation seems excellent for all values of N_0 . We now added the Figure to the supplement of the manuscript.

Figure R9. Analytical approximation dependence on N_0 . **a)** Realisations of equation ?? for different values of N_0 . Here $N_0 = \infty$ corresponds to the approximate expression ?. Note equations ?? and ?? are not normalised. **b)** The normalised equations ?? and ?? are identical even for the smallest possible $N_0 = 1$.

This is now discussed in Lines [468-469] and SI Figure 16 in the revised manuscript.

In line 566, there is a reference to SI Fig. 26b, but I cannot find this figure. I only seem to have SI Figs. 1-12. Am I missing some of your SI figures?

We are sorry. This is a mistake from a previous iteration that we overlooked. This has now been corrected.

In line 588, do you mean $r = i = 30$ rather than $k = i = 30$? On this note, perhaps the index of summation i could be replaced by n for clarity in Eqs. (4), (6), and (14-18).

Thanks for spotting this error, this is corrected now.

We are running short of indices in our notation. We used n before as the total number of cell divisions. It probably is better to stick to i as index in this case.

Reviewers' comments:

Reviewer #1 (Remarks to the Author):

We thank the authors for the thorough rebuttal and feel like all comments have been satisfyingly answered.

Reviewer #2 (Remarks to the Author):

Thank you to the authors for addressing the majority of my initial concerns. They have provided some additional evidence that scaling their stochastic coalescent model to deterministic (rather than stochastic) growth still allows for satisfying fits to the exact numerical simulations, and they have discussed the limitations of their approach in lines 484-491. The authors have clarified that their inferred survival rates in cancer are much higher than has been previously reported because they simultaneously infer a higher mutation rate than has been previously assumed. SI Figures 14-21 are a helpful addition.

Only a few of my concerns remain:

Mutational distance has now been defined by the authors as the number of mutations different between two ancestral cells, although it remains unclear to me whether implicit in this definition is that one of these ancestral cells descended from the other. If so, it would be helpful to clearly state this assumption. If not, then I remain concerned that equation (7) only accounts for the mutations that accumulated along one of the two lineages that separate these cells, which would be inconsistent with the definition of mutational distance. Which pairs of cells are used to calculate the distributions in Figure 2, for example?

Equation (3) (equation 2 in the original version) is described as the distribution of “the number of non-branching events m ,” but I believe this distribution should be $P(m|r) = \binom{r+m-1}{r-1} \beta^r (1-\beta)^m$. Perhaps the authors meant to give the distribution of the total number of divisions n , which I believe should be $P(n|r) = \binom{n-1}{r-1} \beta^r (1-\beta)^{(n-r)}$.

Line 228 states that “cell birth and death on a 2- or 3-dimensional grid was simulated using a Gillespie algorithm.” Each time a plot from the spatial model is given, it would be useful to identify whether it was generated from a 2-dimensional simulation or a 3-dimensional simulation.

The x-axis of the center plot in Figure 4 reads “inferred mutational distance,” but should it instead read “inferred mutation rate” as it does in Figure 3?

The caption for SI Figure 18b reads “(b) Accepted values of the MCMC for the first 1000 steps” but it appears to be the log-likelihood for the first 250 steps. Was the wrong plot included here?

Reviewers' comments:

We want to thank both reviewers again for carefully reviewing our work and providing such constructive feedback to help us improve the manuscript.

Reviewer #1 (Remarks to the Author):

We thank the authors for the thorough rebuttal and feel like all comments have been satisfyingly answered

We want to thank the reviewer again for the very constructive and helpful comments and are very happy to see that we could answer all comments satisfyingly.

Reviewer #2 (Remarks to the Author):

We want to thank the reviewer again for the very constructive assessment of our work that helped us improving our manuscript tremendously. Please find a brief respond to the remaining suggestions below.

Thank you to the authors for addressing the majority of my initial concerns. They have provided some additional evidence that scaling their stochastic coalescent model to deterministic (rather than stochastic) growth still allows for satisfying fits to the exact numerical simulations, and they have discussed the limitations of their approach in lines 484-491. The authors have clarified that their inferred survival rates in cancer are much higher than has been previously reported because they simultaneously infer a higher mutation rate than has been previously assumed. SI Figures 14-21 are a helpful addition.

We are very happy to see that the majority of suggestions have been addressed by our initial revision.

Only a few of my concerns remain:

Mutational distance has now been defined by the authors as the number of mutations different between two ancestral cells, although it remains unclear to me whether implicit in this definition is that one of these ancestral cells descended from the other. If so, it would be helpful to clearly state this assumption. If not, then I remain concerned that equation (7) only accounts for the mutations that accumulated along one of the two lineages that separate these cells, which would be inconsistent with the definition of mutational distance. Which pairs of cells are used to calculate the distributions in Figure 2, for example?

We apologize, if the concept of mutational distances still remained unclear. We were indeed imprecise in how we specified the set-theoretical definition. To clarify: if we take two samples from a simulated tumour, the clonal mutations in those samples correspond to the mutations carried by the ancestral cells of each of these samples. Say these mutational lists are called A and B . By definition, the ancestral cells with mutational profile A and B must coalesce from some earlier ancestral cell C , with mutational profile $C = A \cap B$. We now can construct two mutational distances, namely $y_1 = |A \setminus (A \cap B)|$ and $y_2 = |B \setminus (A \cap B)|$. This process can be iterated for increasing number of samples per tumour (or simulation). We have corrected our statement in the main text of the revised manuscript and apologize again for the confusion caused by our to lose original explanation.

Equation (3) (equation 2 in the original version) is described as the distribution of “the number of non-branching events m ,” but I believe this distribution should be $P(m|r) = \binom{r+m-1}{r-1} \beta^r (1-\beta)^m$. Perhaps the authors meant to give the distribution of the total number of divisions n , which I believe should be $P(n|r) = \binom{n-1}{r-1} \beta^r (1-\beta)^{(n-r)}$.

The reviewer is correct. This was poorly stated in the previous iteration and has now been corrected. We state equation (3) now as the “number of non-branching events m ” now.

Line 228 states that “cell birth and death on a 2- or 3-dimensional grid was simulated using a Gillespie algorithm.” Each time a plot from the spatial model is given, it would be useful to identify whether it was generated from a 2-dimensional simulation or a 3-dimensional simulation.

We agree and thank the reviewer for this suggestion. We have updated the sections of the manuscript correspondingly.

The x-axis of the center plot in Figure 4 reads “inferred mutational distance,” but should it instead read “inferred mutation rate” as it does in Figure 3?

We thank the reviewer for spotting this mistake. This has been corrected now.

The caption for SI Figure 18b reads “(b) Accepted values of the MCMC for the first 1000 steps” but it appears to be the log-likelihood for the first 250 steps. Was the wrong plot included here?

We apologize for being unclear in our caption here. Panel a) shows the entire MCMC for 1000 steps, panel b) shows the accepted values of the 1000 MCMC steps shown in panel a). Approximately 220 suggested parameters were accepted during the realisation of the MCMC chain. We have now clarified the description in the caption of SI Figure 18.